# Prevalence of *POC5* Coding Variants in French-Canadian and British AIS Cohort

**DOI:** 10.3390/genes12071032

**Published:** 2021-07-01

**Authors:** Hélène Mathieu, Aurélia Spataru, José Antonio Aragon-Martin, Anne Child, Soraya Barchi, Carole Fortin, Stefan Parent, Florina Moldovan

**Affiliations:** 1CHU Sainte-Justine Research Center, Montreal, QC H3T1C5, Canada; helene.mathieu.1@umontreal.ca (H.M.); spataruaurelia@gmail.com (A.S.); soraya.barchi.hsj@ssss.gouv.qc.ca (S.B.); stefan.parent@umontreal.ca (S.P.); 2National Heart and Lung Institute (NHLI), Imperial College, Guy Scadding Building, London SW3 6LY, UK; j.aragon-martin@imperial.ac.uk; 3Marfan Trust, National Heart and Lung Institute (NHLI), Imperial College, Guy Scadding Building, London SW3 6LY, UK; annechildgenetics@outlook.com; 4École de Réadaptation, Faculté de Médecine, Université de Montréal, Montreal, QC H3C 3J7, Canada; carole.fortin@umontreal.ca; 5Centre de Recherche, CHU Sainte-Justine, Montreal, QC H1T1C5, Canada; 6Faculty of Surgery, Université de Montréal, Montreal, QC H3T 1J4, Canada; 7Faculty of Dentistry, Université de Montréal, Montreal, QC H3T 1J4, Canada

**Keywords:** *POC5*, adolescent idiopathic scoliosis, cilia, genetics, spine deformity

## Abstract

Adolescent idiopathic scoliosis (AIS) is a complex common disorder of multifactorial etiology defined by a deviation of the spine in three dimensions that affects approximately 2% to 4% of adolescents. Risk factors include other affected family members, suggesting a genetic component to the disease. The *POC5* gene was identified as one of the first ciliary candidate genes for AIS, as three variants were identified in large families with multiple members affected with idiopathic scoliosis. To assess the prevalence of p.(A429V), p.(A446T), and p.(A455P) *POC5* variants in patients with AIS, we used next-generation sequencing in our cohort of French-Canadian and British families and sporadic cases. Our study highlighted a prevalence of 13% for *POC5* variants, 7.5% for p.(A429V), and 6.4% for p.(A446T). These results suggest a higher prevalence of the aforementioned *POC5* coding variants in patients with AIS compared to the general population.

## 1. Introduction

Adolescent idiopathic scoliosis (AIS) is a common disorder characterized by a combination of deviations of the spine in the sagittal and the coronal plane, with vertebral rotation. It affects approximatively 3% of the adolescent population [1,2]; affects females more than males, with a ratio ranging from 1.5:1 to 3:1 [2,3]; and is more prevalent in northern latitudes [4]. The etiology of AIS remains not fully understood, but it is now widely accepted that this disorder has a genetic component, as supported by family history, and higher concordance rates for monozygotic twins compared to dizygotic twins [5,6,7]. Approximatively 40% of AIS patients have a family history [8,9]. The genetic model for AIS remains unclear; indeed, several studies have suggested that it is a polygenic and multifactorial disease [9]. However, other analyses suggest mendelian inheritance, such as autosomal dominant or sex-related, could show with incomplete penetrance [10,11,12]. Since the advent of next-generation sequencing, candidate-gene analysis using pedigrees and population-based genome-wide association studies (GWAS) have been widely used to assess the genetic etiology of AIS. Despite all these efforts, only a few of the candidate genes have been functionally linked to the development of AIS. In 2015, Patten et al. [13] performed a linkage analysis followed by exome sequencing, and identified coding variants in the centrosomal protein gene *POC5* (NM_001099271) in a multiplex four-generation AIS French family [13]. Indeed, a rare SNP, p.(A446T), was found to perfectly segregate with AIS in four families from a pool of 41 AIS French families. Two additional rare variants, p.(A429V) and p.(A455P), were found in the *POC5* gene in AIS sporadic cases. Moreover, all three *POC5* coding variants were functionally related to AIS using a zebrafish model [13]. More recently, a fourth SNP (rs6892146) was identified to be associated with AIS development in a Chinese population [14], but the three other SNPs were not found in this cohort. POC5 is a centriolar protein that is essential for cell cycle progression, cilia elongation [15], centriole elongation, and maturation. Since the identification of the *POC5* gene, a ciliary gene that is strongly associated with AIS, the ciliary pathway has been thoroughly investigated and has revealed promising results [12,15,16,17,18].

To investigate the prevalence of *POC5* genetic variants in AIS, French-Canadian and British AIS patients were screened by targeted or whole-exome sequencing followed by Sanger analyses of DNA.

## 2. Materials and Methods

### 2.1. Patients

One hundred and seventy-seven AIS patients with a Cobb angle of at least 10°, 73 patients from a British cohort (63 unrelated consecutives IS individuals and 10 families), and 104 patients from a French-Canadian cohort (30 unrelated consecutive AIS individuals and 74 patients from 43 families), were recruited. Genomic DNA was extracted from saliva (Cat. RU49000, Norgen, Thorold, ON, Canada) or blood following the protocol provided by the company.

Samples were collected in accordance with the policies regarding the ethical use of human tissues for research. The protocol used in this study was approved by the Centre hospitalier universitaire Sainte-Justine Ethics Committee (#3704).

The control population consisted of an in-house cohort of 1268 individuals with similar ancestry (French, French-Canadian, or European) and was not screened for the presence of AIS [13].

### 2.2. Targeted Exome Sequencing

A library was generated from 10 ng of genomic DNA to perform targeted sequencing of the *POC5* gene using the Ion AmpliSeq (Life Technologies). Sequencing of the 12 exons of *POC5* of 63 British unrelated AIS individuals and 10 families, and 18 French-Canadian families affected with autosomal dominant AIS, was performed by the Centre de Génomique Clinique Pédiatrique intégré CHU Sainte-Justine. The library was prepared using the Ion AmpliSeq DNA and RNA Library Preparation (MAN0006735, Rev. B.0, Ion Torrent, Life Technologies) prior to the exome sequencing following the Ion PGM IC 200 Kit (MAN0007661, Rev. B.0) protocol. Sequencing reads were aligned to the reference human genome sequence (hg19) [19], and the SNPs were identified by Ion Reporter (Ion Torrent). Identified variants were annotated using ANNOVAR [20], which is implemented in the VarAFT software [21] that we used to select exonic and splicing variants with a MAF (minor allele frequency) ≤ 1%. The identified SNPs were then compared to the EVS (Exome Variant Server) database, the Genome Aggregation Database (gnomAD: https://gnomad.broadinstitute.org (accessed on 21 March 2020)), and our in-house control cohort (*n* = 1268).

### 2.3. Whole-Exome Sequencing

A library was generated from 10 ng of genomic DNA to perform whole-exome sequencing using the HiSeq 4000 sequencing machine from the Centre de Génomique Clinique Pédiatrique intégré CHU Sainte-Justine. The gDNA extracted from saliva or blood was sheared to a mean fragment size of 200 pb (Covaris E220 Montreal, PQ, Canada), and gDNA fragments were used for DNA library preparation following the protocol for the SeqCap EZ HyperCap (Roche-NimbleGen, Pleasanton, CA, USA). Enriched DNA fragments were sequenced with 100 pb paired-end reads (HiSeq 4000, Illumina, Vancouver, BC, Canada). Sequencing reads were converted to FASTQ using bcl2fastq software (Illumina) and trimmed by Trimmomatic. The reads were then aligned to the reference human genome sequence (hg19) using the Burrows–Wheeler transform (BWT), followed by a local alignment for indels using Genome Analysis ToolKit software (Broad Institute). Duplicate sequencing reads were excluded by Picard software, and SNPs were identified using the GATK Unified Genotyper and annotated by ANNOVAR software [20], implemented in the VarAFT software [21] that we used to select exonic and splicing variants with a MAF (minor allele frequency) ≤ 1% in the *POC5* gene. The identified SNPs were then compared to the EVS (Exome Variant Server) database, the Genome Aggregation Database (gnomAD: https://gnomad.broadinstitute.org (accessed on 21 March 2020)), and our in-house control cohort (*n* = 1268).

### 2.4. Validation with Sanger Sequencing

All *POC5* coding variants identified by WES or targeted exome sequencing were validated by Sanger sequencing. The segregation of the coding variants was also completed using Sanger sequencing for the take-out-concerned families studied. PCR amplification was performed using the TransStart FastPfu FLY DNA Polymerase (AP231, Civic Bioscience) following the instructions of the manufacturer with primers FWD-5′-GGACCAAACTTTAGCCAGTATG-3′ and RV-5′-TCTCGATCTCCTGACCTCGT-3′. Sanger sequencing of amplicons was performed on an ABI 3730xl DNA Analyzer (Applied Biosystems, Louisville, KY, USA) at Eurofins Genomics (Louisville, KY, USA).

### 2.5. Statistics

The allelic frequency of the *POC5* coding variants was compared to the control population using a one-tailed Fischer’s exact test. A *p*-value < 0.05 was considered statistically significant.

## 3. Results

### 3.1. Patient Enrolment

Since the identification of *POC5* as a candidate gene for adolescent idiopathic scoliosis and its functional validation, we have analyzed the prevalence of *POC5* coding variants within the AIS population, and also have sought to identify new candidate genes. Ninety AIS French-Canadian families were recruited demonstrating different types of transmission: autosomal dominant or recessive, and 30 AIS sporadic cases. For this study, 43 families were selected. Our French-Canadian cohort was supplemented with the 73 UK AIS patients of our collaborators from London, United Kingdom.

### 3.2. POC5 Variants Prevalence Using Next-Generation Sequencing

Among the 53 French-Canadian and British AIS families and 94 unrelated AIS patients, 177 AIS patients were screened for *POC5* coding variants. The combination of whole-exome sequencing and targeted sequencing by AmpliSeq using a targeted Amplicon chip followed by a confirmation with Sanger sequencing revealed that 13% (*p* ˂ 0.0001, Fisher’s exact test) of AIS patients with or without family history were carrying one of the three variants of *POC5,* previously identified as the first causative gene [13]. Indeed, 11 of them carried the A429V variant (6 families and 5 sporadic cases); i.e., 7.5% (*p* ˂ 0.0001), and 11 were found to carry the A446T variant (2 familial, 6 sporadic cases); i.e., 6.4% (*p* = 0.0052). No patient with the A455P variant was reported (Table 1).

The EVS (Exome Variant Server) database [22] reported a minor allele frequency (MAF) in the European-American population of 1.2% for the p.A429V variant and 1.5% for p.A446T. The Genome Aggregation Database (gnomAD: https://gnomad.broadinstitute.org (accessed on 21 March 2020)) [23] reported a MAF of 1.1% for the p.A429V (rs146984380) variant and 1.6% for p.A446T (rs34678567).

### 3.3. Segragation Analysis of POC5 Coding Variants with AIS

The segregation analysis of the *POC5* coding variants with the disease for the AIS families was then performed using Sanger sequencing (Figure 1). The variant p.(A446T) was found in two families (F62 and F80), and showed a perfect segregation with the disease in family 80. Sadly, DNA was not able for the rest of family 62. For the families that were found to carry the variant p.(A429V)—F02, F18, F37, F57, F58, and F66—the segregation analysis showed incomplete penetrance (Figure 1).

## 4. Discussion

We reported the prevalence of *POC5* gene variants in 13% of AIS patients with or without a family history of this condition; that is, six times more frequent than in our in-house control cohort that matched for ethnicity. Two methods were used for the screening of *POC5* gene in both French-Canadian and British families with AIS. This study confirmed the previously reported data by Patten et al. [13], which evidenced three rare SNPs (p.A446T, p.A429V, and p.A455P) in the *POC5* gene in four families of a pool of 41 AIS French families and 150 IS cases from France. We also confirmed the autosomal dominant transmission pattern of *POC5* coding variants in families with AIS.

Adolescent idiopathic scoliosis is a complex disease with a multifactorial etiology including genetic, environmental, and hormonal factors, but the pathogenesis of this disease remains poorly understood. To decipher the genetic implication in AIS, different approaches were used: association studies and linkage analyses to identify causative genes or genes that may impact AIS susceptibility and/or disease progression. Many association studies were performed using GWAS technology and genome-wide linkage analysis, followed by exome sequencing and highlighted multiple-locus candidate, especially on chromosomes 6, 9, 16, 17 [24], and 19 [25]. *POC5*, locus 5q13.3, was the first unambiguously causative gene that was identified and functionally related to the diseases using a zebrafish (*Danio rerio*) model. The three causative variants of the *POC5* gene (p.A429V, p.A446T, and p.A455P) were found in exon 10, and all three corresponded to the substitution of an alanine to another amino acid, suggesting that the alanine in exon 10 of *POC5* could play an important role in the pathogenesis of AIS. Our study supported the importance of *POC5* variants in the AIS population. In AIS patients from French-Canadian and British families, we found *POC5* variants in 13% of AIS cases, but not all the variants showed a perfect segregation with the disease, highlighting the fact that AIS is a complex disease, and very likely a polygenic disorder. In our cohort of French-Canadian subjects, the previously identified variant in IS cases, namely p.A429V, was the most predominant.

To date, these three variants of *POC5* have not been found in the Chinese population [14]; however, in the cohort of this Chinese study, AIS patients were not screened for their family history. It is important to underline that *POC5* gene variants were first identified in a huge multiplex family in which variants and the disease were transmitted in an autosomal dominant manner through four generations. This was consistent with the hypothesis of the initiating role of POC5 in the development of the familial form of AIS.

POC5, a conserved protein, is essential for centriole assembly, elongation, and cell cycle [15,26]. Recent studies identified POC5 as interacting with POC1B, FAM161A, and centrin-2 to build an inner scaffold with an helicoidal assembly, which provides the structural flexibility and strength to maintain microtubule and ciliary cohesion [27]. It can be hypothesized that those alanines in exon 10 of *POC5* are important for the proper interaction of POC5 with its protein partners, and therefore for the proper ciliary integrity. Several osteogenic pathways are hosted on primary cilia, including Sonic Hedgehog, Wnt, or the calcium-signaling pathway, and could play a part in AIS pathogenesis. More recently, primary cilia have been found to be related to bone-mass reduction through the microtubule (MT) network disorganization caused by the reduction of MT anchorage to the basal body [28]. Altogether, these findings could explain the low bone mineral density observed in AIS patients [29].

## 5. Conclusions

The *POC5* gene was one of the first pieces of the puzzle of the genetic etiology of AIS, and since its identification, other genes coding for components of the primary cilia have been found to be linked to this disease [12,16]. Our study confirmed a higher prevalence of *POC5* variants in patients with AIS compared to the general population, as Patten et al. [13] already reported, and this reinforces that *POC5* plays a role in the pathogenesis of AIS. The functionality of those variants also was previously reported [13], and was related to ciliary functionality [15]. Further investigations are necessary in order to identify additional genes and finally draw complete pathways that relate the primary cilia to AIS. Finding causative genes for AIS and understanding the molecular consequences of these gene variants is necessary to improve the knowledge about this disease, especially by deciphering the genetic involvement.

## Figures and Tables

**Figure 1 genes-12-01032-f001:**
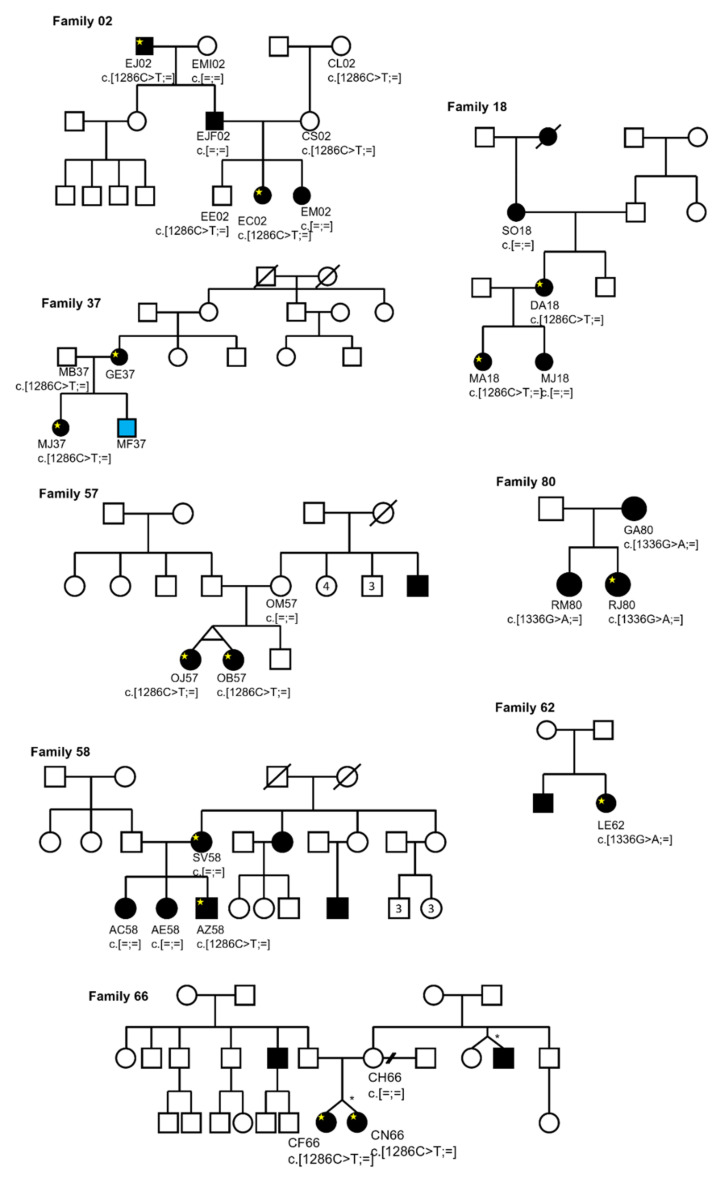
Pedigrees of French-Canadian families showing the co-segregation of POC5 variants (c. 1286C > T (p.(A429V) and c.1336G > A (p.(A446T)) with the disease. Open circles and squares indicate unaffected individuals, Blackened circles and squares indicate affected females and males, respectively. Blue circles and squares indicate juvenile females and males. Yellows stars indicate exomed AIS patients. * Incomplete penetrance.

**Table 1 genes-12-01032-t001:** *POC5* coding variant distribution among the French-Canadian and British AIS cohort compared to 1268 controls. The number of patients from families or sporadic cases that were carrying *POC5* coding variants and the frequency for each of the 2 variants (p.(A446T) and p.(A429V)) are reported.

Data	Families (*n* = 53; 83 AIS Patients)	AIS Cases with Unknown Pedigree data (*n* = 94)	Controls Matched for Ethnicity with Families and Cases (*n* = 1268)	Comparison of Allelic Frequency of the Rare Variants in AIS Cases vs. Controls (Fisher’s Exact Test, One Tailed)
Sequencing Methods	WES + Targeted Exome	WES + Targeted Exome	WES + Sanger	
**p.(A446T)**	2/53 3.8%	6/94 6.4%	19/1268 1.5%	*p* = 0.0052
**p.(A429V)**	6/53 11.3%	5/94 5.3%	9/1268 0.7%	*p* ˂ 0.0001
***POC5* coding variants**	8/53 15.1%	11/94 11.7%	28/1268 2.2%	*p* ˂ 0.0001

## Data Availability

The data presented in this study are available in doi:10.3390/genes12071032.

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
