# Peer review of "Prevalence of POC5 Coding Variants in French-Canadian and British AIS Cohort"

_genes, 2021, doi:10.3390/genes12071032_

Round 1

Reviewer 1 Report

The study is not optimally designed and the conclusions are not supported by the data. 

Introduction: There is much related to the discovery of AIS-associated loci that is lacking in the introduction. As written, the content of the Introduction is mixed between describing Mendelian (not complex) and polygenic (complex) mechanisms underlying AIS genetic risk. The genetic etiologies of these two are not the same, though they are considered equivalent in this study. Familial/Mendelial loci contribute higher disease-causing risk than do modest-effect loci of polygenic complex traits. The Introduction, in presenting the current study, requires descriptions of both of these. It is not apparent that the same mutation can be a risk factor for some but a disease-causing Mendelian allele in others, particularly for these variants (some of which are not actually that rare). The selected references for the epidemiology of AIS are out-dated.

Study design: There are two groups in this study. The first is a cohort of families, for which the mutations do not segregate with disease in several of these. This suggests there is little (if any) relationship between the disease and the mutation. The second cohort is of sporadic cases. The study is flawed in mulitple aspects: 1) Two methods were used for detecting POC5 variants, and the performance of these two approaches to detect POC5 variants is likely different. It would have been improved if all subjects had Sanger sequencing, or at least if all subjects were evaluated with the same NGS method. 2) There are no controls for analysis. 3) Case cohorts are of different ethnic origins and likely can not be compared (also see #2). There are certainly examples of other disease-causing alleles that have higher frequency in the French-Canadian population, though this is unexplored here for POC5.

There is no evidence for association of these variants to AIS in the absence of a control cohort analyzed the same way. The authors report 10 families with POC5 variants in Table 1, which is contributing half of all positive "cases" in the Table. However, in several of these families there are at least one affected individual that does not have any POC5 mutation. So these results are biased. 

Author Response

We thank reviewer 1 for the comments and for the help to improve our work. Please see the attachment for the point-by-point response.

Reviewer 2 Report

  1. Title needs work. “Genetic prevalence” is a meaningless phrase. Title should be something like “Prevalence of POC5 coding variants in French-Canadian and British AIS cohort”.
  2. The frequency of each of the SNPs you discuss needs to be provided for your cohort(s) in unrelated individuals (i.e. one AIS patient per family) and in whatever control cohort you are using. You could probably just use gnomAD or EVS frequencies from non-Finnish Europeans given your populations are both non-Finnish European populations to determine if the frequency of these SNPs is higher than expected. You cite these frequencies but don’t really compare an equivalent frequency estimate for your cohort. Could you add this? Please compare the estimated frequencies of each variant in your unrelated individuals to the frequency in gnomAD for NFE.
  3. What controls are you using in this paper? I can’t find any description of controls in the methods section? Where are p-values coming from? If you compare cases and controls, then you need to say what statistical tests you used in the methods section as well and what samples were used for the tests. I sort of assume you used unaffected family members, but that has its problems too (incomplete penetrance of causal alleles for instance). You state A429V (1.2%), A446T(1.5%), A446V (Not present in EVS), A455P (Not present in EVS).

You have a total of 53 families and 94 unrelated AIS patients. If you use one person per family and each unrelated person, you would have 147 people. Table 1 lists 14 in the row labeled families. It is unclear if that is the number of families with a variant or the number of people with a variant in families or the number of affected people in families with a variant. Can you clarify in the table legend?

  1. Do any of your patients have multiple variants in POC5? Do any of them have other coding variants in POC5 beyond these 3-4 variants?
  2. In figure 1, why don’t you show all of the families that have a segregating variant? You say that there are 14 families but figure 1 only has 8 families. Can you add a supplementary table with each patient or family member and what variant they have and if they have any other variant in POC5? It is not clear where your 14% of cases with a variant comes from.
  3. Do you have a cohort of French-Canadians that have been exome sequenced to determine the frequency of these SNPs in French-Canadians? Maybe there is some population stratification or population bottlenecking that occurred for these variants and the population frequency in French-Canadians is higher than general Europeans. Maybe this is why these variants were found associated in French individuals before?
  4. The intro is not very well written. It has some English issues. Line 44, for instance, states “To find major causative gene, …”. This should be something like, “To find a major causative gene”. Further, it is arguably not a “causative” gene and should not be called such.
  5. In general, you should have someone edit for English mistakes and grammar. I’ve pointed out some mistakes in the minor comments below, but there are many and I am not editing this paper for grammar.

Minor Comments:

  1. You misspell British (Bristish) several times in Table 1.
  2. Line 59 should be “One Hundred” and not just “Hundred”.
  3. Line 62. “Controls Ethical considerations” is not a sentence and is meaningless. Is this supposed to be a header for a section?
  4. line 68. A library was generated, not designed from 10ng of genomic DNA.
  5. line 70. “Consecutives AIS individuals” ?
  6. line 81. A library was generated, not designed from 10ng of genomic DNA.

Author Response

We would like to thank reviewer 2 for the comments and the help to improve our work.

Please see the attachment for the point-by-point response.

Round 2

Reviewer 2 Report

The authors fixed the majority of grammar/English errors and provide adequate explanation of the statistics they used to assess the significance of their associations. I think it is sufficiently improved to accept.